

# Post-traumatic stress symptoms are associated with better performance on a delayed match-to-position task

Meghan D. Caulfield[1,2] and Catherine E. Myers[2,3]

[1] Department of Psychology, Lafayette College, Easton, PA, United States of America
[2] Neurobehavioral Research Laboratory, VA New Jersey Health Care System, East Orange, NJ, United States of America
[3] Department of Pharmacology, Physiology & Neuroscience, Rutgers University-New Jersey Medical School, Newark, NJ, United States of America

## ABSTRACT

Many individuals with posttraumatic stress disorder (PTSD) report experiencing frequent intrusive memories of the original traumatic event (e.g., flashbacks). These memories can be triggered by situations or stimuli that reflect aspects of the trauma and may reflect basic processes in learning and memory, such as generalization. It is possible that, through increased generalization, non-threatening stimuli that once evoked normal memories become associated with traumatic memories. Previous research has reported increased generalization in PTSD, but the role of visual discrimination processes has not been examined. To investigate visual discrimination in PTSD, 143 participants (Veterans and civilians) self-assessed for symptom severity were grouped according to the presence of severe PTSD symptoms (PTSS) vs. few/no symptoms (noPTSS). Participants were given a visual match-to-sample pattern separation task that varied trials by spatial separation (Low, Medium, High) and temporal delays (5, 10, 20, 30 s). Unexpectedly, the PTSS group demonstrated better discrimination performance than the noPTSS group at the most difficult spatial trials (Low spatial separation). Further assessment of accuracy and reaction time using diffusion drift modeling indicated that the better performance by the PTSS group on the hardest trials was not explained by slower reaction times, but rather a faster accumulation of evidence during decision making in conjunction with a reduced threshold, indicating a tendency in the PTSS group to decide quickly rather than waiting for additional evidence to support the decision. This result supports the need for future studies examining the precise role of discrimination and generalization in PTSD, and how these cognitive processes might contribute to expression and maintenance of PTSD symptoms.

## INTRODUCTION

When exposed to a traumatic event, some individuals develop posttraumatic stress disorder (PTSD), including re-experiencing symptoms such as frequent intrusive memories (e.g., flashbacks), believed to be triggered by situations that resemble aspects of the traumatic event (e.g., *Elzinga & Bremner, 2002*; *Brewin & Holmes, 2003*;

Corresponding authors
Meghan D. Caulfield,
caulfiem@lafayette.edu
Catherine E. Myers,
catherine.myers2@va.gov

*Brewin, Gregory & Lipton, 2010*). Learning and memory processes such as generalization and discrimination may underlie re-experiencing symptoms in PTSD. For example, patients with PTSD may be impaired at discrimination, the ability to distinguish and respond differently to similar stimuli, and/or may show increased generalization, the ability to take what is learned about one stimulus and apply it to other. A consequence of increased generalization is that stimuli, which once evoked normal memories, become capable of evoking traumatic memories (e.g., *Tryon, 1998*; *Van Meurs et al., 2014*).

Evidence that PTSD is associated with overgeneralization emerges from differential conditioning studies, in which patients with PTSD often show reduced discrimination between fear and safety cues (*Orr et al., 2000*; *Peri et al., 2000*; *Blechert et al., 2007*; *Jovanovic et al., 2010*). Poor differentiation of fear and safety cues is conceptualized as overgeneralization of fear learning, such that similar fear responses are evoked by both cues. Recent research provides a behavioral link between discrimination and generalization, demonstrating that behavioral pattern separation is related to increased fear generalization in healthy young adults (*Lange et al., 2017*). Given that patients with PTSD demonstrate reduced discrimination learning of fear and safety signals, it is possible that impaired discrimination or generalization may extend to more neutral learning tasks, such as computer-based associative learning tasks. In such tasks, participants are first challenged to learn a series of stimulus-response mappings by trial and error, such as picking the rewarded object from each pair of objects, or learning which of two stimuli should be paired with a target. Later, task demands change by altering irrelevant features of familiar stimuli or by presenting familiar objects in new pairings, making it possible to assess how well participants generalize previously learned associations to the new stimuli or pairings. A body of studies now support the idea that the hippocampal region—although not strictly required for the initial stimulus-response learning—contributes during learning by setting up stimulus representations to support subsequent generalization. Thus, amnesic patients with hippocampal-region damage can learn the initial associations, but then perform at or near chance on the generalization phase (*Myers et al., 2008a*), as do nondemented elderly with hippocampal atrophy consistent with prodromal Alzheimer's disease (*Myers et al., 2002*; *Myers et al., 2003*; *Myers et al., 2008b*). Further, functional imaging studies in healthy adults have shown associations between hippocampal region activity at learning, and successful generalization (*Myers et al., 2008b*; *Shohamy & Wagner, 2008*; *Fera et al., 2014*).

The hippocampus is also implicated in PTSD. Structural neuroimaging studies show reduced hippocampal volume in patients with PTSD (*Gurvits et al., 1996*; *Villarreal et al., 2002*; *Gilbertson et al., 2002*), as well as a relationship between overgeneralization of negative context and bilateral hippocampal volume in PTSD (*Levy-Gigi et al., 2015*). Functional neuroimaging studies suggest reduced or abnormal hippocampal activity in PTSD (*Astur et al., 2006*; *Werner et al., 2009*; *Milad et al., 2009*). Given the reduced hippocampal volume and abnormal hippocampal activity observed in PTSD, it might be expected that Veterans with PTSD would perform similar to patients with medial temporal (hippocampal) amnesia and nondemented elderly with hippocampal region damage: i.e., demonstrating spared learning but reduced generalization on these associative learning tasks. While one prior study did find decreased generalization in patients with PTSD

(*Levy-Gigi et al., 2012*), several other prior studies have now reported the opposite pattern of *increased* generalization by Veterans with clinically-diagnosed PTSD or severe current PTSD symptoms, compared to peers with few/no PTSD symptoms (*Kostek et al., 2014*; *Anastasides et al., 2015*; *Levy-Gigi et al., 2015*) suggesting a more complicated picture of the hippocampus in PTSD.

One explanation is that rather than generalizing, participants with PTSD symptoms may simply be less able to distinguish or discriminate subtle differences between visual stimuli. Poor visual discrimination could promote increased generalization simply because participants fail to detect small differences between visual stimuli. Studies examining visual discrimination in humans frequently use briefly presented images that are very similar (but not identical) to previously presented objects, requiring a high degree of visual discrimination that may reflect multiple underlying processes, such as decreased attention or motivation to learn during the acquisition phase. Discrimination difficulties in such tasks may also reflect a decrease in pattern separation function, a process of comparing new inputs to prior inputs that has consistently been shown to depend on the dentate gyrus of the hippocampus (*Bakker et al., 2008*; *Lacy et al., 2011*; *Reagh & Yassa, 2014*; *Berron et al., 2016*).

To investigate whether decreased discrimination occurs in a visual discrimination task we conducted Experiment 1, which used a sample of Veterans from the same population as several of the earlier generalization studies, along with an additional civilian group. All participants were self-assessed for PTSD symptom severity, as in earlier studies, and grouped according to presence or absence of severe PTSD symptoms (PTSS). Participants were then administered a spatial discrimination task in the form of a visual match-to-sample test, where subjects first saw a target object at a particular screen location, and then after a short temporal delay were challenged to distinguish the target from an identical object presented at a new location. If the PTSS group performed worse on the task, this would suggest that the pattern of increased generalization observed in prior studies might be explained either as a reduction of visual discrimination, or inability to maintain attention and concentration. If no such deficit were found, this would suggest that the increased generalization observed in prior studies is more likely to be due to active processes, such as imbalance in hippocampal or other brain substrates normally mediating the tradeoff between generalization and specificity in learning.

Following Experiment 1, which documented between-group differences, Experiment 2 was conducted to use mathematical analysis techniques in an attempt to better understand possible latent cognitive processes that could differ between groups to produce the observed behavioral pattern.

## EXPERIMENT 1: SPATIAL DISCRIMINATION TASK

To assess spatial pattern separation, a visual delayed match-to-sample task was modified from *Holden et al. (2012)*, which in turn adapted the task for humans based on rat spatial and temporal discrimination paradigms (*Gilbert, Kesner & Lee, 2001*; *Gilbert & Kesner, 2002*). Each trial included a sample phase, in which participants viewed a single circle on the computer screen, followed by a choice phase in which participants identified

which of two circles was in the previously-viewed location. Choice phase stimuli were separated by one of four possible spatial separations: 0 cm, 0.5 cm, 1.0 cm, and 1.5 cm. Smaller spatial separations between the sample phase and choice phase are more difficult, requiring greater pattern separation processes, and presumably greater reliance on the dentate gyrus of the hippocampus. In their study, Holden and colleagues (2012) used this procedure to examine pattern separation processes in cognitively normal young adults and older adults. As expected, when spatial separation increased (and the task became easier) performance improved in both groups. However, older adults did not demonstrate the same degree of benefit of increased spatial separation as younger adults, suggesting less efficient pattern separation processes in older adults and implicating age-related changes in the dentate gyrus of the hippocampus. Holden and colleagues further demonstrated hippocampal contributions in their spatial discrimination task by using a standardized test of verbal memory to compare cognitively unimpaired to cognitively impaired older adult subgroups. This comparison revealed that the cognitively impaired group had significantly poorer performance compared to the unimpaired group, again with the largest group differences at the largest spatial separation (1.5 cm). This result highlights that the task is sensitive to individual differences in spatial discrimination in aging and implicates the hippocampus as playing a possible role in mediating visual spatial separation tasks. The present study extended the use of this task to examine individual differences in spatial discrimination in PTSD.

## METHODS

### Participants

One hundred forty-three participants, including 90 Veterans and 57 civilians (never served in the military), were recruited from the East Orange VA Medical Center and surrounding community, by flyer and word-of-mouth referral. Participants were paid $20/hour for participation in a single 2-hour session. All participants signed written statements of informed consent at the start of the session. Study procedures were approved by the Institutional Review Board at VA NJHCS (Approval #: 01161) and conformed to guidelines established by the Declaration of Helsinki and the Federal Government for the protection of human subjects.

Because the Veteran sample included an overrepresentation of males (about 85% male), we intentionally oversampled females in the civilian sample to adequately power analyses of the effects of gender, as well as PTSS and military service. Additionally, the Veteran group was subdivided *post-hoc* into combat-exposed vs. non-combat-exposed groups (see below for procedure), resulting in three participant groups: civilian, non-combat Veteran, and combat Veteran.

One Veteran participant withdrew from the study before completing all the tasks; this Veteran's data were excluded from subsequent analysis.

The resulting set of 142 participants (87 Veterans and 55 civilians) included 86 males and 56 females, with a mean age of 51.1 years (SD 13.5) and mean education level of 15.0 years (SD 2.7). Asked to self-report race, 96 participants self-identified as Black,

African, or African-American; 23 as White or Caucasian, 11 as Mixed-Race, and 11 as Other (including 1 Native American); 1 subject declined to specify. Asked about ethnicity, 6 participants self-identified as Hispanic, 134 as non-Hispanic, and 2 declined to specify.

Asked about specific conflicts in which they had served, 32 Veterans specified Vietnam, 14 specified Gulf War/Operation Desert Storm, 12 specified Operation Enduring Freedom/Operation Iraqi Freedom, eight specified other conflicts (e.g., Bosnia, Lebanon, Somalia), and 29 reported no specific conflict or peacetime service. Numbers sum to greater than 87 due to some Veterans whose service spanned multiple conflicts.

## Procedures

Testing took place in a quiet testing room, with the participant seated at a table for paper-and-pencil questionnaires and tasks, or at a comfortable viewing distance from a Macintosh i-mac computer (22″ screen) for computer-based tasks. All participants received a battery of questionnaires and tasks, as detailed below. As permitted within the constraints of the 2-hour testing session, some participants also received additional tasks (e.g., piloting of new behavioral tasks or questionnaires); those data are not reported here.

### Questionnaires

All participants received the Posttraumatic Stress Checklist (PCL-C; (*Blanchard et al., 1996*), a 17-question self-report measure of the presence and frequency of PTSD symptoms; symptoms are rated according to how much they have "bothered" the participant in the past month, from not at all (1) to extremely (5). The PCL-C is highly predictive of clinician-rated PTSD in Veterans (*Weathers et al., 1993*; *Blanchard et al., 1996*; *Lunney, Schnurr & Cook, 2014*). Specific PCL-C questions correspond to *DSM-IV* symptom clusters including re-experiencing, avoidance/numbing, and increased arousal. PCL scores of 50+ are predictive of PTSD in military samples (*Weathers et al., 1993*; *Blanchard et al., 1996*). Accordingly, as in prior studies, (*Radell et al., 2017*) we used this cutoff to separate participants according to presence or absence of current, severe PTSD symptoms (PTSS).

Veteran participants also received the Combat Exposure Scale (CES), which assesses exposure to stressful military events (*Keane et al., 1989*). Following prior studies (*Ginsberg et al., 2008*; *Myers, VanMeenen & Servatius, 2012*; *Radell et al., 2017*), Veterans scoring 8 or higher on the CES were classified as having a history of exposure to combat, and those scoring below 8 were exposed as not having combat exposure.

### Trail-Making Test (TMT)

In order to rule out the possibility that any group differences in our spatial discrimination task might reflect differences in motor or cognitive flexibility, all participants completed the trail-making test (TMT), a standard neuropsychological tool to assess visual attention, sequencing, mental flexibility and motor function (*Spreen & Strauss, 1998*). In Part A, participants are presented with a sheet of paper containing 25 encircled numbers, placed in various locations around the page, and use a pencil to "connect the dots" in numerical order as quickly as possible. Part B is similar, except that the page contains 25 encircled numbers and letters, and participants connect the dots alternating letters and numbers (i.e., 1-A-2-B-3-C...) as quickly as possible. If the subject makes an error, the experimenter

calls attention immediately and has the participant proceed from the point where the mistake occurred. Performance is scored as time in seconds to complete each part, with errors counting only in increased time of performance.

### Spatial discrimination task

The spatial discrimination task is a computer-based test of spatial and temporal discrimination adapted from Holden and colleagues (*2012*) and presented on a Macintosh computer with 9 × 12″ screen programmed in the SuperCard development environment (Solutions Etcetera, Pollock Pines CA, USA) and freely available for non-commercial use (https://osf.io/hpek3/). The task involves the presentation of a fixation cross (3 s) located at the center of the screen, followed by a target dot that is offset by a random amount of $0 \pm 3$ cm to the left or right of the fixation cross. Following a variable delay, the participant is presented with a second fixation cross, followed by two dots (the original target dot and a second distractor dot). The participant is asked to identify which dot was the original target by making a key press of "LEFT" or "RIGHT" on a masked keyboard (Fig. 1). Participants can use either or both hands to enter responses but cannot touch the screen or otherwise physically "hold" the target location during the delay period. The experiment consists of three practice phases: one phase to familiarize the participants with the dots task (3.5 cm spatial separation, two trials), a second phase asking participants to read randomly presented numerals (0 to 9; 1.25 cm tall) aloud, and a third phase combining the two so that participants perform the numeral naming task in the delay between target and choice (2 trials). The testing phase is similar in format to the third practice phase, and includes two blocks of 12 trials, including one trial with each of several spatial separations (Low = 0.75 cm, Medium = 1.75 cm, High = 2.75 cm) at each of several delays (5, 10, 20, 30 s). Trial order is randomized within a block, and placement of the distractor location to the left or right of the target is randomized across trials. After every sixth trial, a "rest" screen, appears, offering the subject a chance to take a short break to rest his/her eyes for a few moments, then click the mouse button to resume trials. The entire task took about 15–20 min to complete.

### Data analysis

Statistical analyses were conducted using SPSS version 22 statistical software.

One-way ANOVAs were used to compare the three groups in age and education level, with post-hoc independent-samples t-tests to further explore significant effects. Chi-square test for independence was used to compare male/female distribution and rate of PTSS cases between groups. Based on the results (summarized below and in Table 1), we opted to include gender in all our analyses as a between-subjects factor as there have been observations of sex differences in prevalence of PTSD (*McLean et al., 2011*) as well as learning differences indicating females show enhanced discrimination and generalization during learning (*Day, Reed & Stevenson, 2016*). Since there were no observed differences between groups in education level, this variable was not used as a covariate in subsequent analyses. However, since age is known to affect various speeded-response tasks including the TMT (e.g., *Spreen & Strauss, 1998*), we included age as a covariate in analyses of TMT

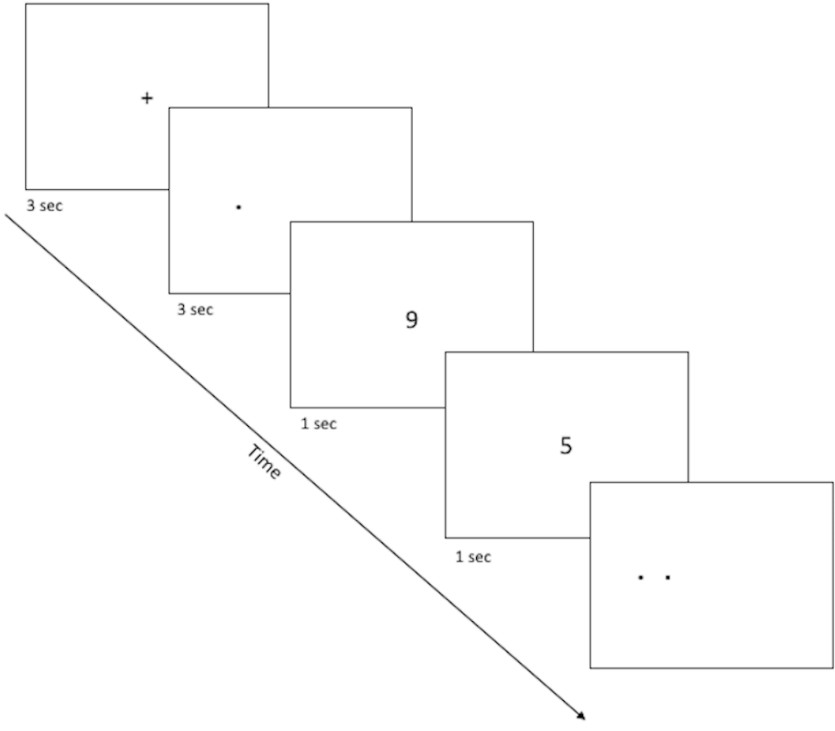

**Figure 1** **Sample screenshots from the discrimination phase of the spatial discrimination task.** A fixation cross appears in the center of the screen for 3 s, followed by a small dot at the target location for that trial. There is a variable delay period, during which the fixation cross re-appears (practice phase 1; not pictured) or else participants complete a digit-naming task, based on numbers that appear for about 1 s each slightly above or below screen center. Finally, dots appear at target and distractor location, and the participant is asked to use the left or right keys to select the dot which appears at the target location (here, the target dot is on the right).

and the spatial discrimination task (after verifying assumption of homogeneity of variance across groups was satisfied).

TMT data were analyzed via mixed ANOVA with dependent measures of time (in seconds) to complete each phase (Parts A and B), between-subjects factors of group (civilian, non-combat Veteran, combat Veteran), gender (male, female), and PTSS (noPTSS, PTSS), and covariate of age.

For the spatial discrimination test, percent correct (averaged across two trials) was scored with each combination of spatial and temporal separation. Mixed ANOVA was then conducted with within-subjects factors of spatial separation (three levels) and temporal separation (4 levels), between-subjects factors of group, gender, and PTSS, and covariate of age.

RT data were also collected for each combination of spatial and temporal separation (averaged across two trials at each combination). The computer program recorded RT in Mac OS ticks (1/60th s), which were converted to ms prior to analysis. Because RT data can be vulnerable to outliers (*Ratcliff, 1993*), data cleansing was performed and any trials with RT < 250 ms or >10 s were dropped from analysis. First, very fast RTs (e.g., <250 ms) can indicate that the subject has (intentionally or unintentionally) pressed the key before onset

**Table 1** Demographic information, PTSD scoring, and TMT performance for the veteran and civilian groups.

|  | Combat-exposed veteran | Non-combat-exposed veteran | Civilian |
| --- | --- | --- | --- |
| N | 33 | 54 | 55 |
| Gender (% female) | 3 female (9.1%) | 13 female (24.1%) | 40 female (72.7%)** |
| Age in years | 49.9 (SD 14.0) | 55.3 (SD 9.9)* | 47.6 (SD 15.4)* |
| Education in years | 15.5 (SD 2.7) | 14.5 (SD 2.2) | 15.1 (SD 3.0) |
| PCL | 56.7 (SD 18.1) | 53.11 (SD 15.9) | 38.2 (SD 14.3)** |
| PTSS cases | 25 PTSS (75.8%) | 33 PTSS (61.1%) | 13 PTSS (23.6%)** |
| TMT Part A (s) | 40.3 (SD 16.4) | 49.8 (SD 22.6) | 48.1 (20.3) |
| TMT Part B (s) | 92.9 (SD 38.9) | 110.3 (SD 48.9) | 101.34 (SD 51.5) |

**Notes.**

One civilian participant declined to specify educational level. Asterisks correspond to independent samples $t$-tests for continuous values, and $\chi^2$ for categorical values. A single asterisk indicates that the non-combat-exposed veteran and civilian groups differed on age (though neither differed from the combat-exposed veteran group). Double asterisk indicates that civilian group differed in distribution of females, PCL score, and rate of PTSS cases, from both veteran groups (which did not differ).

of stimulus presentation; accordingly, trials with such "fast-RT" responses were excluded from RT analysis. Second, especially long RTs are often considered to be the result of other, interfering processes (such as attentional lapse). In healthy young adults given a speeded task, this criterion is often taken to be about 3–5 s. However, in older subjects, psychiatric populations, or individuals on various psychoactive drugs, mean responses are much slower, and the criterion is often extended, to avoid dropping large numbers of trials from analysis (e.g., *Ratcliff, 1993*; *Herzallah et al., 2017*). In the current study, trials with RTs longer than 10 s were excluded, which allowed retention of a large majority of trials while excluding a small number that lay outside the usual range.

After data cleansing, mean RT was computed for correct and incorrect responses at each spatial and temporal separation, and across the task as a whole, for subjects who had at least one trial of the relevant type remaining after data cleansing. Mixed ANOVA was then conducted with within-subjects factors of response (correct vs. incorrect) and spatial separation, between-subjects factors of group, gender, and PTSS, and covariate of age.

Threshold for significance was set at alpha = .05. Where omnibus tests were significant and multiple post-hoc tests were conducted to explore significant results, Bonferonni correction was used to adjust significance thresholds to protect against Type I error; adjusted threshold is only reported where p-values fall below .05 but above the corrected threshold. ANOVA was computed based on Type III SS calculations. Where data failed statistical assumptions for parametric tests (e.g., assumptions of sphericity and equal variance) appropriate corrections were used to adjust degrees of freedom (e.g., Greenhouse-Geisser for ANOVA, Welch's $t$ for $t$-test).

## RESULTS

### Questionnaire results

Based on CES scores, 33 of the 87 Veteran participants were classified as having a history of exposure to combat, and the remaining 54 were classified as non-combat Veterans. Together

with the civilians ($n = 55$), this resulted in three participant groups, as summarized in Table 1. The groups differed significantly in age ($F(2, 139) = 4.80$, $p = .010$, partial $\eta^2 = 0.065$); post-hoc pairwise $t$-tests confirmed that the non-combat Veteran group was significantly older than the civilian group ($t(107) = 3.11$, $p = .002$) but the combat Veteran group did not differ from either the non-combat Veteran group (Welch's $t(51.65) = 1.93$, $p = .059$) or the civilian group ($t(86) = 0.70$, $p = .488$). Given the group differences in age, age was included as a covariate in subsequent analyses. There were no group differences in education ($F(2, 138) = 1.87$, $p = .158$). As expected, the difference in gender distribution was significant ($\chi^2(2, N = 142) = 43.58$, $p < .001$, Cramer's $V = 0.554$); specifically, there were proportionally more females in the civilian group than in the non-combat Veteran group (Yates-corrected $\chi^2(1, N = 109) = 23.91$, $p < .001$, Cramer's $V = 0.487$) or combat Veteran group (Yates-corrected $\chi^2(1, N = 88) = 30.93$, $p < .001$, Cramer's $V = 0.616$) groups, although the two Veteran groups did not differ (Yates-corrected $\chi^2(1, N = 87) = 2.15$, $p = 0.143$,

Table 1 also shows mean PCL scores for each group. ANOVA (with factors of group and gender and covariate of age) confirmed a significant effect of group ($F(2, 135) = 15.45$, $p < .001$, partial $\eta^2 = 0.186$) with no main effect of gender ($F(1, 135) = 0.10$, $p = .757$) and no gender-group interaction ($F(2, 135) = 1.79$, $p = .170$), although the effect of the covariate was significant ($F(1, 135) = 7.39$, $p = .007$). Post-hoc pairwise $t$-tests revealed that the civilian group had lower PCL scores than either the combat Veteran ($t(86) = 5.38$, $p < .001$) or non-combat Veteran group ($t(107) = 5.24$, $p < .001$); the two Veteran groups did not differ ($t(85) = 0.96$, $p = .339$). In Veterans, CES did not correlate with PCL scores (Pearson's $r = .128$, $n = 87$, $p = .238$).

Using the cutoff of PCL-C $\geq 50$ to define PTSS, 71 participants (50.0%) met criteria for PTSS across the three groups (civilian, non-combat, combat). The distribution of PTSS cases across groups differed significantly ($\chi^2(2, N = 142) = 26.72$, $p < .001$, Cramer's $V = 0.434$), with proportionately fewer cases among civilians than Veteran groups (both Yates-corrected $\chi^2 > 14.0$, both $p < .001$); however, rates did not differ between the two Veteran groups (Yates-corrected $\chi^2 = 1.37$, $p = .241$).

### Trail-making test

Unsurprisingly, scores on TMT parts A and B were highly correlated (Pearson's $r = .617$, $p < .001$), with subjects who scored high (poor performance) on part A tending to also score high (poor performance) on part B. However, TMT part A and B scores did not correlate with either CES or PCL (all $p$'s $> .05$). Mixed measures ANOVA with repeated measures factor of TMT part A and part B score with between-subjects factors of group, gender and PTSS status, revealed a significant effect of part ($F(1, 130) = 123.57$, $p < .001$), with subjects generally taking longer to complete part B ($M = 102.7$ s, $SD = 48.0$) than part A ($M = 46.9$ s, $SD = 20.6$), but with no effects of gender, group, or PTSS status and no interactions (all $p$'s $> .070$).

### Spatial discrimination task: percent correct responding

Consistent with previous studies reporting poorer spatial discrimination in aging, our sample also showed a negative correlation between task performance (total percent

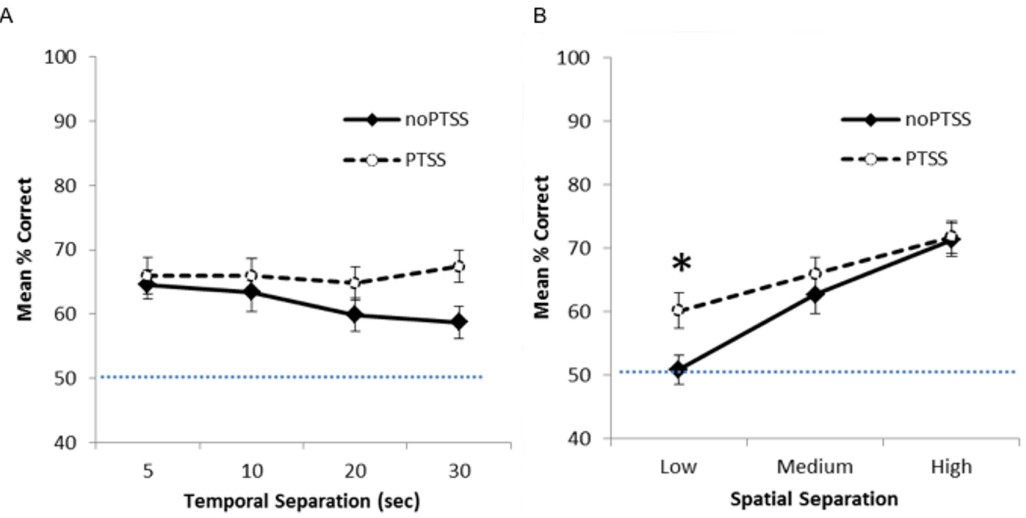

**Figure 2 Spatial discrimination task performance of PTSS and noPTSS groups.** (A) On the spatial discrimination task, there was no main effect of temporal delay on response accuracy; however, performance improved as spatial separation between target and distractor increased. (B) Collapsing across temporal delays, participants meeting PTSD symptom criteria (PTSS group) outperformed those not meeting PTSS criteria (noPTSS), specifically at the lowest (i.e., most difficult) spatial separation. Asterisk indicates significant difference between PTSS and noPTSS groups.

correct) and age, with younger subjects outperforming older (Pearson's $r = -0.227$, $p = .007$, $r^2 = 0.052$). The correlation remained after partialing out effects of group ($r = -0.264$, $p = .002$, $r^2 = 0.070$). Accordingly, age was entered as a covariate in the subsequent analyses of task performance.

Figure 2 shows mean percent correct across the four delays (Fig. 2A) and the three spatial separations (Fig. 2B) for PTSS collapsed across civilian and Veteran groups. As expected, for the noPTSS group, performance was better at high spatial separation than low spatial separation, and better for shorter delay than for longer delay. However, the PTSS group appeared less sensitive to reductions in spatial separation or increasing temporal delay.

Mixed ANOVA with Greenhouse-Geisser correction was used to examine performance on the spatial discrimination task at varying temporal and spatial separations. There was an interaction between PTSS and spatial separation ($F(1.85, 238.01) = 3.35$, $p = .040$, partial $\eta^2 = 0.025$), as well as a significant within-subjects effect of spatial separation (i.e., worse performance as separation decreased, $F(1.85, 238.00) = 5.52$, $p = .005$, partial $\eta^2 = 0.0741$). There was no within group effect observed for temporal separation (i.e., no effect of delay between target and choice, $p = .430$). No other interactions or effects reached significance (all $p$'s > .050), except for the effect of the covariate age ($F(1, 129) = 5.49$, $p = .021$).

To further investigate the interaction between spatial separation and PTSS, data were collapsed over the four temporal separations, as shown in Fig. 2B, and Bonferroni-corrected paired $t$-tests run at each spatial separation. Participants meeting PTSS criteria outperformed noPTSS participants specifically at the low (i.e., most difficult) spatial separation ($t(140) = .003$, Cohen's $d = 0.485$), but not at the medium ($t(140) = 0.90$,

$p = .370$, $d = 0.169$) or high (i.e., easiest) spatial separation ($t(140) = 0.13$, $p = .898$, $d = 0.022$).

The above analyses included all 142 subjects, including some who performed at or near chance in terms of percent correct calculated across all 24 trials. To determine if those who "solved" the task were driving observed differences in the above analyses, we also analyzed the data from only those participants who made at least 17 out of 24 correct responses (70.8% correct), i.e., significantly better than chance based on binomial calculations, $P(X > 16) = .032$). This included 34 of 71 PTSS participants and 22 of 71 noPTSS participants, a difference that fell short of statistical significance (Yates-corrected $\chi^2(1) = 3.57$, $p = .059$). When performance was analyzed in this subset, results look qualitatively similar to those shown for the full dataset in Fig. 2; specifically, the PTSS group continued to outperform the noPTSS group at the low but not medium or high spatial separation (Bonferroni-corrected paired $t$-tests, Low separation: $t(54) = 2.90$, $p = .005$, $d = 0.716$; Medium separation: $t(54) = 0.10$, $p = .922$, $d = 0.042$; High separation: $t(54) = 2.06$, $p = .044$, $d = 0.563$).

Finally, performance at the lowest spatial separation was significantly correlated with PCL scores ($r = .229$, $n = 142$, $p = .006$, $r^2 = 0.052$), although performance at the higher spatial separations was not (Medium: $r = .130$, $p = .122$, $r^2 = 0.169$; High: $r = .064$, $p = .451$, $r^2 = 0.004$). The correlation between PCL scores and performance at the lowest spatial separation held even when considering only the $n = 56$ "solvers" ($r = .396$, $p = .003$, $r^2 = 0.157$). There was no correlation between performance at any spatial separation and CES (all $p$'s > .08).

## Spatial discrimination task: reaction time

During data cleansing for RT analysis, a total of 75 trials (2.2% of total) were excluded as long-RT (RT > 10 s); the vast majority of participants (129 of 142) made 1 or fewer long-RT responses, and only three subjects (including one who made 12 long-RT responses) made more than five long-RT responses. Additionally, eight "fast-RT" trials (RT <250 ms) were excluded: one subject made 5 responses with RT = 0 and two subjects made 1–2 responses with RT = 0. No other RT <250 ms were recorded.

Mean RT data for trials at each spatial separation are shown in Fig. 3A, for trials with correct and incorrect responses. However, many subjects did not have RT scores for all combinations of temporal and spatial separation; in particular, 32 subjects did not make any incorrect responses on any trials at the largest spatial separation. For those 101 subjects where all six mean RT scores were available, mixed ANOVA on mean RT for correct and incorrect responses on trials at each spatial separation showed a within-subjects effect of response, as subjects responded faster on correct than incorrect trials ($F(1, 88) = 11.868$, $p = .001$, partial $\eta^2 = 0.119$); there were also significant three-way interactions between trial type, group, and PTSS ($F(2, 88) = 3.46$, $p = .036$, partial $\eta^2 = 0.073$) and between trial type, PTSS and gender ($F(1, 88) = 4.45$, $p = .037$, partial $\eta^2 = 0.049$); no other between-subjects effects of group, gender, or PTSS, and no interactions, approached significance (all $p$'s > .100) except the interaction between trial type and the covariate age ($F(1, 88) = 6.84$, $p = 0.010$). Post-hoc testing to explore the three-way interactions (i.e., mixed ANOVA to
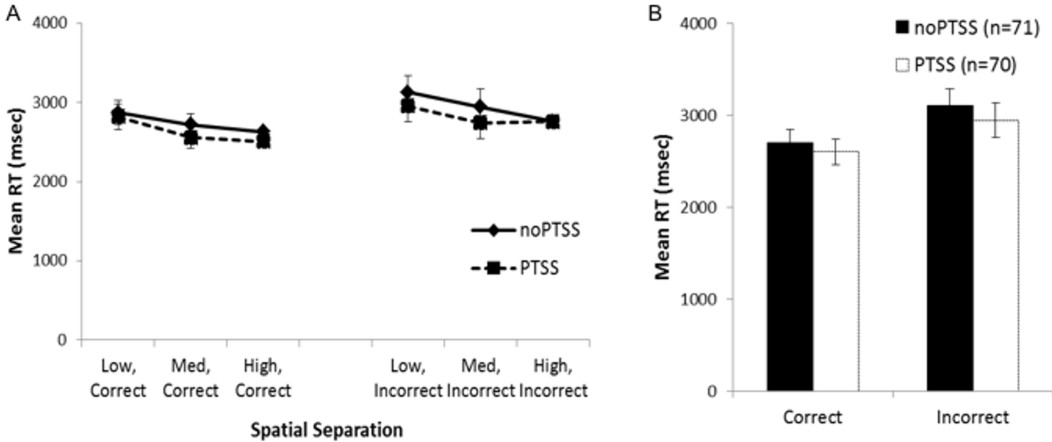

**Figure 3 Response reaction times for the visual discrimination task.** (A) Reaction time (RT) for correct and incorrect responses, on trials at each spatial separation. There was a within-subjects effect of response type (i.e., faster RT for correct than incorrect responses) but no effect of gender or PTSS and no interactions. (B) Analyses were similar when considering mean RT for correct and incorrect trials (regardless of spatial separation), which allowed consideration of all but one subject's data.

explore type-gender and type-group interactions in the PTSS and noPTSS subgroups) did not reveal significant effects (all $p > .050$).

Because the above analysis excluded a number of subjects, a secondary analysis was conducted on the RT scores, collapsing across all spatial separations; allowing mean RT for correct and incorrect trials to be computed for all except a single subject in the PTSS group who made no incorrect responses on the task (Fig. 3B). There was again a within-subjects effect of response type ($F(1, 129) = 4.11$, $p = .045$, partial $\eta^2 = 0.031$); no other effects or interactions approached significance (all $p$'s $> .100$).

In summary, although the RT data showed an expected within-subjects effect of faster responding on correct than incorrect trials, neither analysis showed evidence of a PTSS (or gender) effect on RTs. Because the direction of effect—with the PTSS group showing *better* performance at the most difficult spatial separation, without evidence of slower RT mediating this accuracy—was unexpected, we turned in Experiment 2 to explore latent cognitive processes that could underlie these results.

## EXPERIMENT 2

Two-choice forced-response tasks, such as the spatial discrimination task studied in Experiment 1, generally involve a speed-accuracy tradeoff, whereby better performance can be achieved at the cost of slower RT, or faster RT at the expense of reduced accuracy. Because these two behavioral variables interact, analysis of either alone does not give a complete picture of the underlying cognitive processes mediating performance.

To address this issue, various mathematical models have been developed. Drift diffusion models (DDMs) represent a class of model commonly used to infer latent cognitive processes underlying two-choice decision-making, and to link these processes to neural mechanisms (*Ratcliff, 1978*; *Ratcliff & McKoon, 2008*; *Ratcliff et al., 2016*).

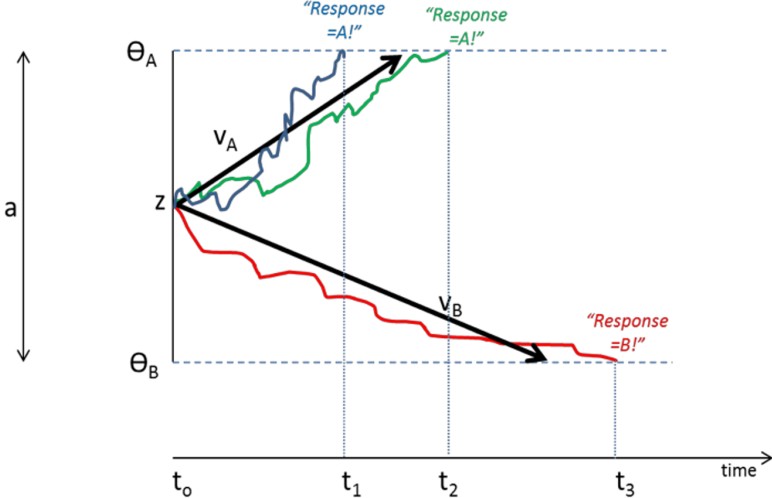

**Figure 4 Schematic of Drift Diffusion Model (DDM) of decision-making.** Each trial starts at time $t_0$; noisy information provides evidence which drives the decision-making process from a starting point ($z$) towards boundaries $\Theta_A$ or $\Theta_B$; when a boundary is reached, the corresponding response A or B is selected. The solid blue line schematizes an example trial in which evidence accumulates across time, eventually reaching criterion for response A at time $t_1$; actual RT for this trial reflects $t_1$ plus nondecision time Ter (time to encode stimuli and execute motor responses). The green line shows another trial in which stochastic processes lead to a slightly longer decision time $t_2$. The red line schematizes a trial in which evidence accumulates favoring response B, reaching criterion at time $t_3$. Drift rates $v_A$ and $v_B$ govern the slope of the decision process with higher drift rates producing faster decision times (and faster RTs). Greater boundary separation ($a = \Theta_A - \Theta_B$) produces "response caution" resulting in greater response accuracy at the expense of longer RTs. Parameters $v_A$, $v_B$, $a$, $z$, and Ter vary across individual subjects but are inferred from observed RT and accuracy data.

DDMs assume that organisms make rapid (often implicit) decisions based on accumulation of noisy information over time. In the case of a two-choice forced response task, as schematized in Fig. 4, the decision process on each trial begins at a starting point "z". Noisy accumulation of information pushes the decision process towards one of two boundaries representing criteria for the two possible responses. When a boundary is reached, the corresponding response is executed. Within-trial variability (noise) means that the same information does not always produce the same RT or even the same behavioral responses.

Central to the DDM approach is the idea that different individuals can be described in terms of different "parameters" that together govern the decision process. These parameters include the rate of information processing (called the drift rate, v), the *a priori* response bias (i.e., starting point z may be closer to one boundary than the other), and nondecision time, Ter, which includes stimulus processing and response execution time. Subjects can also differ in the distance between boundaries (a), a parameter often assumed to encapsulate "response caution" since higher values mean that more evidence must be accumulated before a boundary is reached and a response selected. Later elaborations of the DDM have included additional variables encoding intertrial variability, although these parameters may increase model complexity without necessarily increasing explanatory power (*Lerche & Voss, 2016*).

Several methods have been proposed for fitting the DDM to individual subject data, i.e., using a participant's observed RT and accuracy measurements to infer (or "recover") parameters such as v, Ter, a, and z that best describe that participant's observed trial-by-trial behavior (*Ratcliff & Tuerlinckx, 2002*; *Ratcliff & McKoon, 2008*; *Wiecki, Sofer & Frank, 2013*; *Voss, Voss & Lerche, 2015*); these methods—while mathematically complex—are available in computationally tractable form (e.g., "Fast-dm-30" of *Voss, Voss & Lerche, 2015*). However, many of these methods require at least 10 trials with each possible outcome; this may require a large number of trials if accuracy is high (so that erroneous responses are rare). A simplified method (the "EZ" model of *Wagenmakers, Van Der Maas & Grasman, 2007*), later extended to the "EZ2" or "robustEZ" model in (*Grasman, Wagenmakers & Van der Maas, 2009*) uses an approach similar to signal detection theory to estimate DDM parameters using only mean, variance, and accuracy of correct RTs.

DDMs have been used successfully in cognitive neuroscience, to help elucidate differences in cognitive processing between control and patient populations, and to link behavior with neurophysiological data (e.g., functional neuroimaging) to show how brain systems encode and integrate sensory information and use it to generate behavioral responses. An important application of the approach is that such latent cognitive processes may suggest targets for therapy, in patient populations with impaired decision-making. For example, *Herzallah et al. (2017)* used DDM models to show that patients with Parkinson's disease (PD) have higher response caution a (thus requiring more evidence before making responses), and patients with major depressive disorder (MDD) display reduced drift rate for stimuli associated with positive (put not negative) feedback, while patients with comorbid PD and MDD display both features. The DDM analysis therefore allowed inference of underlying features of the cognitive deficit in these patient groups that were not evident from analyzing observed behavior alone.

We adopted a similar approach here, applying DDM analysis to individual subject data from Experiment 1, and then testing for group differences in extracted parameter values for subjects in the PTSS vs. noPTSS groups. Experiment 1 revealed that the PTSS group performed better than the noPTSS group in the Low spatial separation (most difficult) trials. Most interestingly, this improved performance did not come with an increase in response time, suggesting that the better performance could not simply be attributed to a speed/accuracy tradeoff, with PTSS participants responding more slowly and carefully. In this analysis, our goal was an initial exploration of latent cognitive mechanisms that could produce the observed behavior in which the PTSS group outperformed the noPTSS group at the Low spatial separation, but not at the other, easier separations—without a concomitant increase in RT.

## METHODS

Because the behavioral task involved a small number of trials (24 per subject), divided among multiple experimental conditions, analysis via "full" DDM was inappropriate, since this requires a minimum of about 10 trials with each outcome (correct and incorrect) for each condition. Instead, we used the "robust-EZ" DDM method (*Wagenmakers et al., 2008*)

which bases analysis on RT for correct trials only and which can be used with a fairly small number of trials. To do this, we grouped trials by spatial separation, resulting in eight trials per spatial condition. Further, because many subjects performed at or near ceiling on the High (easy) spatial separation, we focused on the Low (difficult) and Medium spatial separation conditions. The robust-EZ DDM method requires calculating mean and variance of reaction time for trials with correct responses (cMRT, cVRT), as well as proportion of trials with incorrect responses (proportion error trials, Pe), so we calculated these summary statistics for each subject based on trials at the Low and Medium spatial separation. Since the robust-EZ method requires Pe>0, we were forced to drop data from 16 subjects who had 100% correct responses at the Medium spatial separation ($n = 12$), the Low spatial separation ($n = 1$), or both ($n = 3$); eight further subjects were excluded who had <2 correct responses at the Low ($n = 6$) or Medium ($n = 2$) spatial separation, since RT variance could not be calculated. This left 118 participants for whom all six summary statistics were available. There were no significant differences between these 118 participants and the remaining 25 subjects in gender, group, or PTSS status (chi-square test with Yates correction for $2 \times 2$ tables, all $\chi^2 < 4.0$, all $p > .140$), in age ($t(140) = 1.23$, $p = .223$) or in task performance including total percent correct, RT on correct trials, or RT on incorrect trials (all $t < 2.0$, all $p > 0.05$).

The six summary variables for each subject were provided to the "robust-EZ" DDM model (*Wagenmakers et al., 2008*), available as the EZ2 package for R (https://rdrr.io/rforge/EZ2/). Specific R commands used are provided in the Appendix. The model was used to estimate values of five free parameters for each subject: an overall value of a (boundary separation), z (starting point), and Ter (nondecision time), as well as separate drift rates v27 and v63 for trials involving low spatial separation (27 cm) and medium spatial separation (63 cm). These estimated values for each subject are those which provide the best fit to the subject's trial-by-trial data, in terms of minimizing error between the model predictions and the subject's actual behavior (response and RT) across trials.

After obtaining estimated parameter values (v27, v63, z, a, Ter) for each subject, these parameter values were then used as dependent values analyzed via ANOVA, with between-subject factors of gender, group, and PTSS status. For assessing model goodness-of-fit, EZ2 output provides a "value" score for each subject, which is sum of squared prediction error (SSE) multiplied by $10^7$ for precision; these were converted back to SSE and univariate ANOVA was used to compare goodness of model fit as a function of gender, group, and PTSS status.

## RESULTS

The model converged for all subjects except one (a 28-year-old male civilian; his data were excluded from further analysis). SSE averaged 0.049 (*SD* 0.037, range 0.004–0.234). There were no differences in goodness-of-fit measure for the model as a function of subject group or PTSS status (univariate ANOVA, all $p > .100$ except group $p = .072$). Univariate ANOVAs were conducted on each of the five parameters, with factors of gender, group, and PTSS status. For v27 (drift diffusion rate, Low spatial separation), PTSS participants

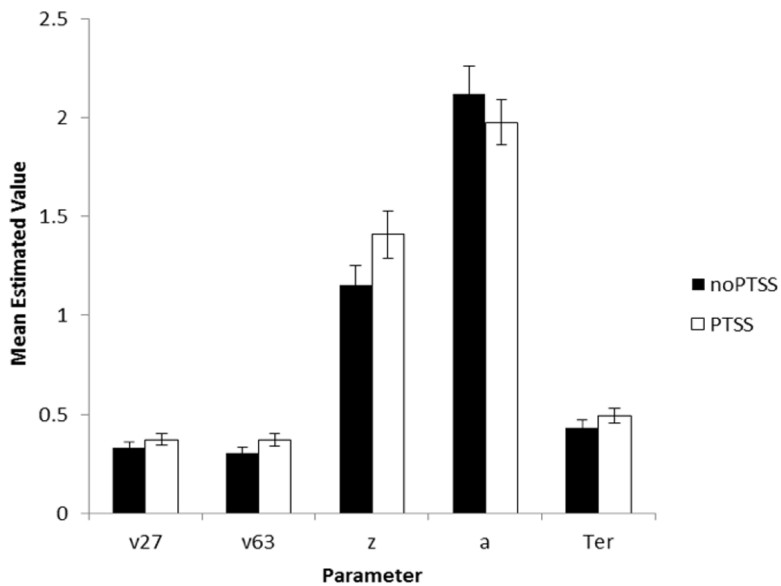

**Figure 5** **Estimated parameters, based on drift diffusion model (DDM), for PTSS and noPTSS groups.**
There were modest but significant effects of PTSS status on two parameters: v27 (drift diffusion rate at the
Low spatial separation) and a (response caution). Specifically, the PTSS group had higher drift diffusion
rate for trials at the Low spatial separation (v27), indicating faster accumulation of evidence, and lower
values of response caution (a), indicating less evidence was required before deciding to execute a response.
Other parameters including drift diffusion for trials at the Medium spatial separation (v63), starting point
(z), and nondecision time (Ter), did not differ between groups.

had higher values than noPTSS subjects ($F(1, 105) = 5.90$, $p = .017$, $\eta^2 = 0.053$), with no
effects of group or gender and no interactions (all $p > .100$). PTSS participants also had
higher response caution a ($F(1, 105) = 4.30$, $p = .041$, $\eta^2 = 0.039$), with no effects of group
or gender and no interactions (all $p's > .070$). There were no effects of group, gender or
PTSS status on v63 (drift diffusion rate, Medium spatial separation), starting point $z$, or
nondecision time Ter (all $p > .100$). Figure 5 summarizes estimated parameter values, for
PTSS and noPTSS groups. Specifically, the PTSS group had higher drift diffusion rate for
trials at the Low spatial separation (v27), indicating faster accumulation of evidence, and
lower values of response caution (a), indicating less evidence was required before deciding
to execute a response. Other parameters including drift diffusion for trials at the Medium
spatial separation (v63), starting point (z), and nondecision time (Ter), did not differ
between groups.

In summary, Experiment 2 used DDM modeling to further examine decision making
processes in the spatial separation task given the unexpected result in Experiment 1,
where PTSS participants outperformed noPTSS participants in the hardest (Low spatial
separation) trials with no slowing of RT. DDM modeling suggested two mechanisms that
contributed to performance of the PTSS and noPTSS groups. First, PTSS participants
had greater drift diffusion rate on trials with Low spatial separation, indicating faster
accumulation of evidence on the more difficult trials (but not on the easier trials at Medium
spatial separation). Second, PTSS participants had lower response caution, indicating that

less evidence was required before selecting a response. This bias for impulsivity in the PTSS group would be exacerbated by a bias against further accumulation of evidence on the more difficult trials.

## GENERAL DISCUSSION

To determine if poor visual discrimination underlies increased generalization previously observed in Veterans with PTSD we used a visual match-to-sample pattern separation task. Given prior research documenting increased generalization in PTSD, it might be expected that the PTSS group would show correspondingly decreased discrimination, particularly on the more difficult trials. Instead, current results showed that the PTSS group outperformed the noPTSS group on the hardest trials of visual spatial discrimination task, i.e., those trials with the smallest amount of spatial separation.

A simple explanation for this result would have been in terms of a speed/accuracy tradeoff: the PTSS group might have simply increased their accuracy by taking more time to carefully evaluate and respond to stimuli—which would be reflected by slower RT in the PTSS group. However, this was not the case; if anything, the PTSS group responded slightly faster than the noPTSS group. Thus, slower RT cannot explain better discrimination by the PTSS group. To determine which mechanisms might lead to better discrimination without increase in RT, we turned to computational modeling using a DDM model. The DDM model indicated that the PTSS group both accumulated evidence faster (higher drift rate $v$) and have lower response caution (reduced boundary separation $a$ between possible responses, indicating reduced response caution). This result indicates that PTSS participants may tend to be impulsive in their decision making and also accumulate evidence faster in selecting a decision; the latter in particular could promote faster RTs without necessarily entailing a reduction in performance. Future research directly addressing decision-making processes in PTSS is warranted to understand the differential contribution of these processes and their role in PTSS.

It is also possible that hippocampal differences previously observed in PTSD are not entirely at odds with the present findings. First, it is important to stress that neuroimaging was not available for participants in the current study; therefore, hippocampal volume reductions could not be verified. A further possibility is that the present task is only relying on a portion of the hippocampal structure. It may be that while the entire hippocampus is smaller, the functions of individual substructures may be altered in different ways. Support for this notion comes from other visual discrimination tasks in healthy participants which have found that pattern separation is more precisely related to the dentate gyrus of the hippocampus (*Bakker et al., 2008*; *Berron et al., 2016*). The role of the dentate gyrus in spatial pattern separation is further supported by extensive work using rat models. *Gilbert, Kesner & Lee (2001)* used a similar spatial delayed-match-to-sample paradigm as we used in our experiment. Using a cheeseboard maze apparatus they required rats to uncover a baited food well (i.e., sample phase), and then following a delay select between one of two wells (i.e., choice phase): one well was the same as the sample phase (correct choice) and a second was varying distances from the sample phase (incorrect choice). Rats with dentate gyrus

lesions had impaired performance as the small spatial separations (more difficult trials) but improved as spatial separations increased and the task became easier. Given the similarities between the two tasks, it would be expected that the present task is also reliant specifically on the dentate gyrus, suggesting that differences in dentate gyrus activity may underlie better discrimination performance in the PTSS group. However, given data suggesting that hippocampal volume reduction is a pre-existing risk factor for PTSD (*Gilbertson et al., 2002*), rather than emerging in the wake of PTSD or trauma exposure, it is possible that differences in dentate gyrus structural or functional activity may also be a pre-existing risk factor, leading to similar distribution of dentate gyrus impairments among our PTSS and noPTSS participants. Future research using spatial visual discrimination tasks in fMRI to elucidate the contribution of hippocampal substructures in healthy, aging, and PTSD participants is clearly warranted.

Another consideration is the type of pattern separation task. Many fMRI studies use pattern separation tasks that ask participants to discriminate highly similar stimuli (*Bakker et al., 2008*; *Lacy et al., 2011*; *Berron et al., 2016*), rather than discriminate identical stimuli at different spatial locations (as in the present task). To our knowledge, no study has included both types of task to determine if performance on these tasks is correlated. Further work should examine whether differences observed in discrimination processes in PTSD/PTSS participants in a spatial separation task would carry into other tasks of pattern separation.

It is also possible that our pattern separation task was tapping into other cognitive domains that were not necessarily measured or controlled in the present study. Participants in our study were permitted to respond with one or both hands on the keyboard. Therefore, it cannot be ruled out that inhibitory processes or lateralization influences, such as the Simon effect, were playing a role in performance differences. Previous research examining response inhibition in PTSD proposes that a reliance of bottom-up processing and a breakdown of inhibition results in hyperarousal and poor attentional control (*Bryant et al., 2005*). While our task was one of spatial discrimination, it cannot be denied that it involved a considerable amount of attention and decision-making processes. Furthermore, poor executive control or response inhibition may also have interacted with participants' handedness preferences. Recent research utilizing a go/no go task measuring inhibitory processes reported that not only were patients with PTSD more likely to make inhibition-related errors than participants without trauma, but the process of inhibition relied on different brain networks such that controls relied on a right-lateralized cortical network while the PTSD participants activated only on the left-lateral frontal cortex during inhibition (*Falconer et al., 2008*). Additionally, recent research examining the relationship between fear generalization and behavioral pattern separation reports brain areas involved in inhibition and safety processing play a role in behavioral pattern separation including the subcallosal cortex, orbitofrontal cortex, and ventromedial prefrontal cortex (*Lange et al., 2017*). This is in line with recent work using rat models of PTSD which further implicate the prefrontal cortex as a key region leading to deficits in cognitive flexibility and inhibitory responses (*George et al., 2015*). Future research more explicitly examining

inhibition, decision making, and its relationship to performance in pattern separation tasks in PTSD is necessary.

Finally, there are a number of other factors which may affect cognitive function, such as socioeconomic status and intelligence, which were not controlled for in the current study, but could be examined in future work. Likewise, future studies could examine depressive symptoms and other symptoms which are often comorbid with PTSD, which could be contributing to the current results; in particular, depressed patients have sometimes shown overgeneral memory (e.g., *Watkins & Teasdale, 2001*), while a prevalent hypothesis of depression suggests that depression is associated with reduced levels of neurogenesis in the dentate gyrus, potentially affecting pattern separation processes in the dentate gyrus (for review, *Samuels & Hen, 2011*). Therefore, it would be very interesting to examine this spatial discrimination task in patients with depression alone or in combination with PTSD symptoms.

Notwithstanding these limitations, our results provided no evidence to suggest that increased generalization in participants with PTSS is due to reduced ability to discriminate. In fact, results indicate the opposite, with PTSS participants showing better discrimination performance. It is important to consider that our expectations for the current study were derived from prior work suggesting that participants with PTSS/PTSD show increased generalization on computer-based associative learning tasks. However, the present task was more precisely a task of discrimination. While generalization and discrimination are usually considered to be (negatively) related, they are not necessarily opponent processes, and as such may depend on different hippocampal substructures (*Bakker et al., 2008*; *Rolls, 2013*). Therefore, future research using both generalization and discrimination tasks in the same sample of participants is necessary to determine if the same participants who show heightened generalization are those that also show heightened discrimination.

## ACKNOWLEDGEMENTS

Opinions stated herein do not necessarily represent the official views of the Department of Veterans Affairs or the US Government. For assistance with subject recruitment and data collection, the authors wish to thank Yasheca Ebanks-Williams. The spatial discrimination task used in the current study was adapted from a version developed by Ray Kesner and colleagues (unpublished), whom the authors wish to thank for helpful discussions during software development and task piloting. An executable version of the current spatial discrimination task is available as open access software (freely available for non-commercial use) on Open Source Framework (https://osf.io/hpek3/).

### Funding

This work was supported by Merit Review Award I01 CX000771 from the US Department of Veterans Affairs Clinical Sciences Research and Development Service, and by the NSF/NIH Collaborative Research in Computational Neuroscience (CRCNS) program and NIAAA

R01 AA018737. The funders had no role in study design, data collection and analysis, decision to publish, or preparation of the manuscript.

### Grant Disclosures

The following grant information was disclosed by the authors:
US Department of Veterans Affairs Clinical Sciences Research and Development Service: Merit Review Award I01 CX000771.
NSF/NIH Collaborative Research in Computational Neuroscience (CRCNS) program.
NIAAA R01 AA018737.

### Competing Interests

Catherine E. Myers is an Academic Editor for PeerJ.

### Author Contributions

- Meghan D. Caulfield analyzed the data, prepared figures and/or tables, authored or reviewed drafts of the paper, approved the final draft.
- Catherine E. Myers conceived and designed the experiments, performed the experiments, analyzed the data, contributed reagents/materials/analysis tools, prepared figures and/or tables, authored or reviewed drafts of the paper, approved the final draft.

### Human Ethics

The following information was supplied relating to ethical approvals (i.e., approving body and any reference numbers):

All participants signed statements of informed consent at the start of the session. Study procedures were approved by the Institutional Review Board at VA NJHCS and conformed to guidelines established by the Declaration of Helsinki and the Federal Government for the protection of human subjects.

### Data Availability

Because these data were collected from Veterans who provided informed consent based on the assurance (following VA Privacy and Information Security Policy) that their individual data would not be made public, we cannot provide the raw data in a publicly-available form. The raw data are archived within the secure VA computing environment, and researchers interested in access to (de-identified) data can contact the authors regarding a possible Data Use Agreement. The software implementing the generalization task described in the paper is freely available for non-commercial use on the Open Source Framework (https://osf.io/hpek3/), and the details of DDM analysis (R code) are available in a Supplemental File.

### Supplemental Information

Supplemental information for this article can be found online at http://dx.doi.org/10.7717/peerj.4701#supplemental-information.

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
