# Peer review of "Post-traumatic stress symptoms are associated with better performance on a delayed match-to-position task"

_PeerJ, doi:10.7717/peerj.4701_

## Round 0.1 · original submission · Major Revisions

I have received comments from two reviewers. You can see that both reviewers have positive impressions about your study but also have some reservations on how you present it. Therefore, I'd like to invite you to rewrite your manuscript according to the reviewers' comments. Please resubmit your revised manuscript along with a point by point reply to the reviewers' comments.

Tsung-Min Hung, Ph.D.
PeerJ editor
Distinguished professor
Department of Physical Education
National Taiwan Normal University

Reviewer 1 ·

Basic reporting

no comment

Experimental design

no comment

Validity of the findings

no comment

Additional comments

The current study examined the association of PTSD with performance in spatial and temporal discrimination task. Three groups of participants were recruited, civilian, non-combat veteran, and combat veteran, based on their symptom severity. The key findings were that individuals with PTSS had higher accuracy rate in the spatial discrimination task without compromised prolonged RT. To further explore the possible accounts to the unexpected performance of individuals with PTSS, the authors conducted a second experiment using the Drift Diffusion Model (DDM), and realized that individuals with PTSS could accumulate evidence faster and have lower response caution during decision making. Generally speaking, the current manuscript is well-written and can provide insightful information to the decision-making process of PTSD. However, several caveats should be taken care of by the authors before being published on the PeeJ.

Introduction-
1. The authors are recommended to reorganize the paragraphs regarding behavioral and neurobiological correlates of PTSD and non-PTSD. Maybe they can address the behavioral findings first (e.g., findings in associative learning task), and then go to the neurobiological aspects (i.e., findings in hippocampus) in another paragraph. Currently, the authors address this topic interchangeably between behavioral and neurobiological findings, which make the flow difficult to follow.

Method-
2. The major shortcoming of the current manuscript is the use of non-clinical assessment for PTSD. The authors are strongly recommended to justify the use of these self-assessed checklist and scale considering that their findings are basically inconsistent with that from others.
3. The authors are suggested to justify the examination for cognitive flexibility as well as the use of the TMT.
4. The description of the spatial discrimination task should be consolidated. The current version is somewhat lengthy.
5. Also, the authors should provide additional information for why they included “gender” as a factor into data analysis. Are they seeking for a gender-moderating effect?

Experiment 1 & 2 Discussions-
6. Discussions for experiment 1 & 2 should be consolidated to shorter length, and leave most of the major discussions to the General Discussion.

General Discussion-
7. See 6.
8. Generally, the discussion are easy-to-follow except that the authors are suggested to provide some explanation/mechanism to why the PTSS group outperformed the non-PTSS group

Minor issues-
9. The scheme of the spatial discrimination task in Figure 1 should be revised. The graphic in the current version is not clear enough.

Reviewer 2 ·

Basic reporting

Authors of the current study examined visual discrimination using a pattern separation task in subjects with PTSD symptoms (PTSS). Subjects with PTSS showed better discrimination than matched healthy controls which was not linked to slower response time. The authors argue that this links to previously reported PTSD-related overgeneralization associated with hippocampal abnormalities. Finally, the authors used a diffusion model to examine the task accuracy and RT. They found a faster rate of evidence accumulation and lower threshold for the PTSS group.

Based on previous work by the authors, subjects with PTSS showed enhanced generalization with higher levels of PTSD symptoms (re-experiencing). In the current study, however, the authors present results in the exact opposite direction where subjects with PTSS are better at differentiation (discrimination), which is the opposite of compression (generalization). It would be very useful if the authors can provide a theoretical framework to mend the gap between these different empirical findings. For example, the author's aforementioned work on generalization in PTSS is not sufficiently discussed in the introduction or discussion to explain the difference between impaired generalization, decreased generalization, increased generalization, and over-generalization.

Finally, the authors ought to consider the various studies on visual discrimination in human and animal models of PTSD that can significantly improve the current arguments.

Experimental design

There could be an important confound in the task if the Simon effect is controlled for where learning is better for congruent (on the same side) rather than noncongruent stimulus-response associations.

Did the authors examine correlations between questionnaires and task results to look for potential covariates to include the ANOVA model?

The authors report exclusion of various trials with long or short RTs. What was the adopted criterion for this exclusion?

In the modeling section, how was the model fits performed? Which parameters were varied according to trial type and which ones were held constant? A bit more detail in this section would help the reader understand the modeling approach better.

Validity of the findings

Although the findings present an interesting new direction in this field of research, the current results still require further investigation to clarify the variables in action. The results are robust and the analytical approaches are sound. It would be very useful if the authors can add a measure of effect size.

The main issue with the current study is linking the current findings with previous literature on PTSD and generalization. Further, there is a wealth of human and animal literature on pattern separation and the hippocampus that can improve to the introduction/discussion sections.

---

## Round 0.2 · Minor Revisions

I have now received two reviewers’ comment and both reviewers were generally satisfied with your reply and revisions from previous comments. However, one reviewer has pointed out some minor issues that require your additional attention. Please address these issues and provide a point by point reply in addition to the revised manuscript.

Tsung-Min Hung, Ph.D.
PeerJ editor
Distinguished professor
Department of Physical Education
National Taiwan Normal University

Reviewer 1 ·

Basic reporting

Minor issue:
Line 43- should be "respond differently" rather than "respond different"?

Data analysis-
For clarification, please add texts elaborating the statistics for grouop homogeneity (chi-square test and t-tests), as shown in Table 1.
Also, though the authors did not found group difference in demographic measures such as gender ratio or education, texts elaborating control for these factors if group homogeneity was not met (e.g., descriptions to ANCOVA) are needed.

Experimental design

N/A

Validity of the findings

Just come to mind, there should be some factors which may affect cognitive function (e.g., socioeconomic status, intelligence) but were not adjusted for in the current study. This issue should be briefly discussed. Likewise, there may be comorbidity relating to PTSD (e.g., depressive symptoms) which were not controlled for in the current study. The authors should consider these limitations and elaborate how these limitations can be mitigated in the current study.

Additional comments

The authors did a good job in revising the manuscript. I am satisfied with most of the modifications made to the manuscript except several minor issue left to the authors for consideration.

Reviewer 2 ·

Basic reporting

No further comments.

Experimental design

No further comments.

Validity of the findings

No further comments.

Additional comments

The authors addressed all points raised initially. I have no further comments. This is a very interesting paper that will present a valuable addition to the PTSD literature.

---

## Round 0.3 · accepted · Accept

I have read through your reply to the reviewer's comment and your revised manuscript. I am satisfied with your response and decided that there is no need to send to the reviewer. You and your coauthors have my congratulations. Thank you for choosing PeerJ as a venue for publishing your research work and I look forward to receiving more of your work in the future.

Tsung-Min Hung, Ph.D.
PeerJ editor
Research chair professor,
Department of Physical Education,
National Taiwan Normal University

#